# Dynamics of SARS-CoV-2 Variants of Concern in Vaccination Model City in the State of Sao Paulo, Brazil

**DOI:** 10.3390/v14102148

**Published:** 2022-09-29

**Authors:** Svetoslav Nanev Slavov, Debora Glenda Lima de La-Roque, Pericles Natan Mendes da Costa, Evandra Strazza Rodrigues, Elaine Vieira Santos, Josiane Serrano Borges, Mariane Evaristo, Juliana de Matos Maçonetto, Adriana Aparecida Marques, Jonathan Milhomens, Felipe Augusto Rós, Vagner Fonseca, Alex Ranieri Jerônimo Lima, Gabriela Ribeiro, Loyze Paola Oliveira de Lima, Pedro Manuel Marques Garibaldi, Natasha Nicos Ferreira, Glenda Renata Moraes, Elaine Cristina Marqueze, Claudia Renata dos Santos Barros, Antonio Jorge Martins, Luiz Lehmann Coutinho, Rodrigo Tocantins Calado, Marcos Borges, Maria Carolina Elias, Sandra Coccuzzo Sampaio, Marta Giovanetti, Luiz Carlos Junior Alcantara, Dimas Tadeu Covas, Simone Kashima

**Affiliations:** 1Center for Cell-based Therapy (CTC), Blood Center of Ribeirão Preto, Tenente Catão Roxo Street, 2 501, Ribeirão Preto 14051-060, Brazil; 2Butantan Institute, Vital Brasil Avenue, 1500, São Paulo 05503-900, Brazil; 3Pan-American Health Organization/World Health Organization, Lote 19—Avenida das Nações, SEN—Asa Norte, Brasília 70312-970, Brazil; 4Serrana State Hospital, Ribeirão Preto Medical School, University of São Paulo, Nossa Senhora das Dores Street, 811, Serrana 14150-000, Brazil; 5Health Department, Tancredo Almeida Neves Street, 176, Serrana 14150-000, Brazil; 6College of Agriculture “Luiz de Queiroz”, University of São Paulo, Pádua Dias Street, 235, Piracicaba 13418-900, Brazil; 7Oswaldo Cruz Foundation, Rio de Janeiro 4365, Manguinhos, Rio de Janeiro 21040-900, Brazil; 8Department of Science and Technology for Humans and the Environment, University of Campus Bio-Medico di Roma, 00128 Rome, Italy

**Keywords:** SARS-CoV-2 variants, variants of concern (VOC), Gamma, Delta, Omicron, COVID-19 vaccines

## Abstract

From a country with one of the highest SARS-CoV-2 morbidity and mortality rates, Brazil has implemented one of the most successful vaccination programs. Brazil’s first model city vaccination program was performed by the CoronaVac vaccine (Sinovac Biotech) in the town of Serrana, São Paulo State. To evaluate the vaccination effect on the SARS-CoV-2 molecular dynamics and clinical outcomes, we performed SARS-CoV-2 molecular surveillance on 4375 complete genomes obtained between June 2020 and April 2022 in this location. This study included the period between the initial SARS-CoV-2 introduction and during the vaccination process. We observed that the SARS-CoV-2 substitution dynamics in Serrana followed the viral molecular epidemiology in Brazil, including the initial identification of the ancestral lineages (B.1.1.28 and B.1.1.33) and epidemic waves of variants of concern (VOC) including the Gamma, Delta, and, more recently, Omicron. Most probably, as a result of the immunization campaign, the mortality during the Gamma and Delta VOC was significantly reduced compared to the rest of Brazil, which was also related to lower morbidity. Our phylogenetic analysis revealed the evolutionary history of the SARS-CoV-2 in this location and showed that multiple introduction events have occurred over time. The evaluation of the COVID-19 clinical outcome revealed that most cases were mild (88.9%, 98.1%, 99.1% to Gamma, Delta, and Omicron, respectively) regardless of the infecting VOC. In conclusion, we observed that vaccination was responsible for reducing the death toll rate and related COVID-19 morbidity, especially during the gamma and Delta VOC; however, it does not prevent the rapid substitution rate and morbidity of the Omicron VOC.

## 1. Introduction

The SARS-CoV-2 pandemic dramatically hit Brazil and the country became a world leader in COVID-19-related morbidity and mortality (30.4 million cases with 663,000 deaths until April 2022) [1]. However, by March 2021, Brazil undertook a massive COVID-19 vaccination campaign culminating in one of the highest percentages of vaccinated individuals (79.6%) worldwide. The large territory of the country contributed to the entry of different variants of concern (VOCs), leading to massive epidemic waves. In addition, important VOCs, such as the Gamma variant [2], emerged in Brazil. The application of mass vaccinations led to a diminution in the Delta epidemic wave compared to other countries but was ineffective towards the Omicron VOC [3].

The role of pioneer model city for the studies related to the effects of the vaccination was performed by the town of Serrana, where, for the first time in Brazil, between February and April 2021 the CoronaVac vaccine (Sinovac Biotech) was applied, before the beginning of the official vaccination program in the country. This clinical study related to mass vaccination in adults (>18 years old) was named Project S (Portuguese version, Projeto S). This test of the effectiveness of the CoronaVac vaccine demonstrated a dramatic drop in COVID-19 mortality as deaths were reduced by more than 95% [4].

To comprehensively evaluate the extent of the circulating SARS-CoV-2 variants and their substitution dynamics before and after the vaccination program, we undertook a large-scale sequencing program aimed at investigating in real-time all of the SARS-CoV-2 positive samples obtained in the city of Serrana (between June 2020 and March 2022). Beyond the molecular surveillance, we also examined the patients’ clinical outcomes (symptomatology and lethal outcome) plotted against the VOC distribution in the timeline.

## 2. Materials and Methods

### 2.1. Ethical Statement and Study Location

This study was approved by the Institutional Ethics Committee of the Medical School of Ribeirão Preto (Process CAAE: 50367721.7.1001.5440). The study was performed in the town of Serrana located at latitude 21°12′41″ south and longitude 47°35′44″ west in the state of São Paulo. Its population is estimated to be 46,166 inhabitants.

### 2.2. Molecular Confirmation of SARS-CoV-2 Infection

Viral RNA was automatically extracted from 100 uL of nasopharyngeal swab suspension (Extracta kit AN viral, Loccus) in extractor Extracta 32 (Loccus) following the manufacturer’s guidelines. The SARS-CoV-2-RT-PCR was executed using the Gene FinderTM COVID19 Plus RealAmp kit (OSang Healthcare Co. Ltd., Gyeonggi-do, Korea), which detects fragments of the RdRp, E, and N genes. The reaction was performed according to the manufacturer’s protocol using a 7500 Real Time PCR cycler (Thermo Fisher Scientific, Waltham, MA, USA).

### 2.3. SARS-CoV-2 Sequencing

For the genomic sequencing, 60–89% of all SARS-CoV-2 positive samples from each epidemiological week (epiweek) were included. The main criteria for sample selection were the cycle threshold (Ct) value under 35 for at least two of the tested viral genes.

SARS-CoV-2 complete genome sequences were obtained using the COVIDSeq kit (Illumina, San Diego, CA, USA) according to the manufacturer’s protocol. Sequencing libraries were pooled, normalized to 4 nM, and denatured with 0.2 N NaOH and 400 mM Tris-HCL (pH-8). The final sample library (9 pM) was loaded into a MiSeq Nano Reagent Kit v2 (300 cycles), and run on an Illumina MiSeq sequencer (Illumina, San Diego, CA, USA).

### 2.4. Bioinformatics Pipeline

The raw sequence data obtained were submitted to quality control analysis using the FastQC software, version 0.11.8 [5]. Trimming was performed using Trimmomatic version 0.3.9 [6] to select high-quality sequences. The sequences with quality scores > 30 were used. We mapped the trimmed sequences against the SARS-CoV-2 Wuhan reference genome (Genbank refseq NC_045512.2) using BWA [7] and SAMtools [8] software for read indexing. The mapped files were submitted to refinement with Pilon [9] to obtain the indels and insertions in the most correct way possible. Afterwards, the trimmed sequences were subjected to a remap against the genome refined by Pilon. Finally, we used BCFtools [10] for variant calling, and seqtk [11] to create the consensus genomes. Positions covered by fewer than 10 reads (DP < 10) and bases of quality lower than 30 were considered a coverage gap and converted to Ns. Coverage values for each genome were calculated using SAMtools v1.12 [8]. SARS-CoV-2 lineages were assigned by pangolin v.4.1.1.

### 2.5. Phylogenetics Analysis

The phylogenetic tree is composed of 7283 genomes: 4375 genomes from Serrana town and 2908 global representative SARS-CoV-2 genomes [12] that were downloaded as a representative dataset from GISAID (https://gisaid.org/ accessed on 21 May 2022). Sequence alignment was performed using MAFFT v7.475 [13] and manually curated to remove artifacts using Aliview [14]. Maximum likelihood (ML) phylogenetic trees were constructed using Iqtree 2 [15] with the statistical support of ultrafast bootstrap applying 1000 replicates. The nucleotide substitution model was GTR + G4 + F, chosen according to the BIC (Bayesian Information Criterion) statistical model. The final formatting and visualization of the phylogenetic tree were performed using the ggtree R package [16].

### 2.6. Statistical Analysis

For statistical analysis, the software GraphPad Prism version 6.01 was used. We applied multiple comparisons using Kruskal–Wallis with Dunn’s test for and α = 0.05, *p* > 0.05 (not significant), *p* < 0.05 (*), *p* < 0.01 (**), *p* < 0.001 (***), and *p* < 0.0001 (****). 

## 3. Results

### 3.1. Aspects of the Tested Population and Performed Sequencing

During the routine SARS-CoV-2 diagnosis, we obtained 4538 SARS-CoV-2 genomes in the period between June 2020 and April 2022, comprising three subsequent COVID-19 waves in Brazil. Of them, we analyzed 4375 (96.4%) sequences that presented high-quality parameters including a mean number of reads (986,784), mean coverage (98.5%), and mean depth, i.e., 5610. The analyzed genomes were distributed by VOC like this: 1653 Delta cases (37.8%), 1053 Gamma cases (24.1%), 1513 Omicron cases (34.6%), 75 Zeta cases (1.7%), and 81 cases of other lineages (1.9%). The patients who were included in this study were distributed like this: 55.1% were female and 44.9% were male with a mean age of 38 (±18) years. Although the participants were of all ages, most were found in the group between 21 and 50 years old in all of the three epidemic waves (Figure 1), whereas 11.6% were pediatric patients under 18 years of age. Among the identified VOCs and VOIs, so far, we identified 52 SARS-CoV-2 sub lineages in the studied location. The clinical outcome of COVID-19 in the tested patients was evaluated using the WHO clinical score (Figure 2). Most of the cases were characterized as mild (88.9%, 98.1%, 99.1% to Gamma, Delta, and Omicron, respectively) regardless of the VOC in question. Delta VOC mild cases were mostly identified in the population group aged under 40 years old (Figure 2). For the moderate cases, the Gamma VOC were mostly found in the population group aged between 40–50 year old, while the Delta VOC affected individuals below 40 years of age. Interestingly, the moderate Omicron cases affected elderly people.

#### 3.1.1. Introduction of SARS-CoV-2 in Serrana and First Epidemic Wave (February 2020–November 2020)

In Brazil, the first COVID-19 wave is estimated to have been between the epidemiological weeks 9 and 45, 2020. By this time, the performed genomic surveillance showed the presence of parental SARS-CoV-2 lineages coinciding with the epidemiological situation worldwide. We obtained 45 complete SARS-CoV-2 genomes during this epidemic wave which belonged to B.1.1.33 (55.6%), B.1.1 (42.2%), and P.2 (2.2%), respectively (Figure 3a). Most of the SARS-CoV-2 positive patients were aged between 41–50 years of age and generally presented with mild symptoms (54.5%), but 36.4% showed moderate symptoms, and in 9.1% of the positive patients, the symptoms were classified as severe (Figure 3b). 

#### 3.1.2. The Second Epidemic Wave (November 2020–October 2021) Related to the Gamma and Delta Variants of Concern

The second epidemic wave in Serrana occurred between the 46th epiweek (2020) and the 39th epiweek (2021). This period was related to the progressive growth of the sequenced genomes, raising from 7% to 100% of all positive cases per epiweek. By the end of October 2021, we detected 28 SARS-CoV-2 sublineages: 81.9% Gamma VOC (P.1, P.1.11, P.1.12, P.1.14, P.1.15, P.1.2, P.1.7, and P.1.8); 9.4% Delta VOC (AY.100, AY.25, AY.26, AY.3, AY.34.1.1, AY.36, AY.43, AY.43.2, AY.46.3, AY.99.1, and AY.99.2); 5.8% Zeta VOI (P.2), and 2.9% other lineages (B.1.1, B.1.1.28, B.1.1.519, B.1.1.7, C.37, N.9, P.4, and P.7) (Figure 4a).

The second epidemic wave was related to the emergence and rapid dissemination of Gamma VOC in Brazil, and in the town of Serrana. The first Gamma VOC case appeared in the 5th epiweek (February 2021). The complete substitution of all lineages by Gamma VOC took 11 epiweeks (5th–16th epiweeks), and Gamma predominance was observed until epiweek 34 (Figure 5). We continued to detect single Gamma VOC strains until October 2021, when we observed a complete disappearance of this VOC after the detection of only four cases (3%) in epiweek 42 (Figure 5). Alpha VOCs (3/1 272; 0.2%) appeared only during epiweeks 9 and 15. 

The first Delta VOC cases in Serrana appeared in the 31st epiweek (August 2021). Cases started to grow exponentially from the 33rd epiweek until the complete substitution of Gamma VOC by Delta VOC in epiweek 43. The length of time for the replacement of these two VOCs was similar (Figure 5). However, while the Gamma wave was characterized by P.1 and its sublineages (66.4% P.1, 10.5% P.1.7, 2.6% P.1.14, 1.5% P.1.12, and 0.9% of P.1.11, P.1.15, P.1.2, and P.1.8), the Delta wave was comprised of different Delta lineages (3.7% AY.3, 2.0% AY.99.2, 1.2% AY.26, 0.9% AY.34.1.1, 0.7% AY.46.3, and 0.9% of AY.100, AY.25, AY.36, AY.43, AY.43.2, and AY.99.1). The last Delta VOC strain detected in Serrana city occurred in epiweek 3 (2022), followed by the complete substitution of this VOC by the Omicron VOC (third epidemic wave).

Most of the patients infected during the second epidemic wave were aged between 31 and 41 years of age. Compared to the first epidemic wave, the percentage of patients with mild symptoms during the second epidemic wave was considerably higher (89.8%). However, the second epidemic wave coincided with the mass vaccination campaign in Serrana town, with the first dose on 17 February 2021, followed by a second dose 4 weeks later. From the first Gamma VOC case (epiweek 5) to the complete vaccination of all individuals in the city (epiweek 15), 71.4% were mild cases, 19.1% were moderate, and 9.5% were classified as severe. After the complete vaccination of the city until the end of the second epidemic wave (epiweeks 16–42), the percentage of mild cases rose to 94.5% while the moderate and severe cases dropped to 3.9% and 1.5%, respectively.

#### 3.1.3. Third Epidemic Wave (December 2021–April 2022)

In Brazil, the third epidemic wave started at the 51st epiweek (2021) and continues to date. Interestingly, while in Brazil there was a notable reduction in COVID-19 cases from October to December, in Serrana we observed an increasing number of cases (Figure 5, green line). Between the second and the third epidemic wave in Serrana town we identified the circulation of Delta VOC strains such as AY.99.2 (71.2%), AY.3 (18.7%), AY.26 (3.3%), and others (AY.100, AY.101, AY.122, AY.25, AY.34.1.1, AY.43, AY.43.1, AY.43.2, AY.46.3, AY.5, AY.6, and AY.99.1). 

The third epidemic wave coincided with the end of the year the intensive dispersions of humans and the emergence of the Omicron VOC. During this period, an exponential rise in the number of COVID-19 positive cases was observed, reaching three times greater than the previous epidemic waves—especially the Delta VOC wave (Figure 1), but also the Gamma emergence in Brazil. By the 50th epiweek of 2021, we identified the first Omicron VOC in Serrana city. Omicron VOC disseminated so rapidly that in five weeks it completely replaced the Delta VOC. This was a record when compared to other waves such as the Gamma and Delta replacements. 

To date, we identified 16 Omicron VOC strains during the third epidemic wave: BA.1 (23.2%), BA.1.1 (18.1%), BA.1.9 (15.0%), BA.1.14 (13.4%), BA.1.15 (10.6%), BA.1.14.1 (7,9%), BA.2 (2.7%), and others (BA.1.1.16, BA.1.17, BA.1.18, BA.1.13, BA.1.1.1, BA.1.17.2, BA.1.5, BA.1.1.14, BA.1.14.2) (Figure 6a). During this period, we also identified several remaining Delta VOC strains, such as AY.99.2 (4.8%), AY.3 (0.6%), and AY.34.1.1 (0.2%). Most of the infected individuals were female (57.9%) and were aged between 21 and 30 years of age (Figure 6b). Despite this epidemic wave being characterized by the highest numbers of infected individuals, the total number of deaths was greatly reduced compared to previous waves, especially the Gamma VOC shown in Figure 5 by the red line. Figure 5 shows overall the introduction of SARS-CoV-2 to Serrana during the epidemic waves (from February 2020–July 2022) and thereafter.

### 3.2. Global SARS-CoV-2 Phylogenetic Analysis in the Town of Serrana

In this study, we performed a phylogenetic analysis of all of the SARS-CoV-2 genomes that were obtained from the city of Serrana. The utilized dataset contained 2908 reference sequences retrieved up to April 2022 from GISAID sampling worldwide. The obtained global phylogenetic tree showed well-defined clusters which corresponded to the main VOCs circulating in this region. As observed in Figure 7, the largest clusters corresponded to the epidemic waves described above, Gamma, Delta, and Omicron. The new strains generated in this study were randomly interspersed within the reference SARS-CoV-2 strains. For example, considering the collection date, Delta cluster was represented by multiple introductions events in this region. This is explained by the fact that the multiple introductions of the main SARS-CoV-2 lineages occurred over time in these regions, which supports the epidemiologic events observed in the SARS-CoV-2 pandemic not only in the state of São Paulo but also in Brazil, and worldwide.

## 4. Discussion

This study investigated the SARS-CoV-2 molecular dynamics between the period of 2020–2022 in a model vaccination town in Brazil (Serrana) that has gained international importance [4]. The first confirmed COVID-19 case in this city dates back to April 2020 [17], almost two months (25 February 2020) after the first confirmed SARS-CoV-2 case in Brazil [18], reflecting the intensive dissemination speed of SARS-CoV-2 in the state of São Paulo. During the initial period of the COVID-19 pandemic, Serrana followed the pre-established measures by the Ministry of Health of Brazil including isolation, lockdowns, reduction in public transportation, similar to the worldwide guidelines [19]. Further, under the Butantan Institute’s guidance, Serrana participated in the first stepped-wedge trial to assess the efficiency of the CoronaVac vaccine. Briefly, this clinical trial was conducted between the epidemiological weeks 6 and 19 of 2021, coinciding with the mid-Gamma VOC wave of the second wave when the Gamma VOC was introduced and became predominant (for more information of the campaign organization, refer to reference [20]). Additionally, by this time São Paulo state had established the Network for Pandemic Alert of SARS-CoV-2 variants, that improved the genomic surveillance in this region and helped to understand the introduction routes of the SARS-CoV-2 variants in the town of Serrana. 

During the ongoing vaccination program, we observed a significant gap between the vaccination process and the lack of molecular data concerning the circulating SARS-CoV-2 variants; thus, we estimated the effect of the vaccine on the SARS-CoV-2 diversity and the clinical impact that was measured by the percentage of observed lethal and severe cases, respectively. With a focus on this, we adopted a large-scale sequencing survey on the circulating SARS-CoV-2 variants which was aimed at the sequencing of all of the positive cases in this location. By this way, we estimated the SARS-CoV-2 substitution dynamics which followed the trends of emergence and VOCs’ dissemination in Brazil from the introduction of the parenteral strains in the region [21], and the emergence of the first epidemic Gamma VOC wave [22]. This profile was also observed for the Delta epidemic wave [23]. The performed genomic surveillance also helped to identify some rare VOIs, such as C.37 which was circulating in the Andean countries but was underrepresented in Brazil, and occurred mostly as introductions [24,25]. This shows the importance of the systemic monitoring of the circulating SARS-CoV-2 variants that gives important molecular epidemiological information, helping not only to monitor rare lineages but also to predict the introduction of important VOCs and to predict further epidemic waves. 

We believe that the application of the vaccine in Serrana town was related to reduced morbidity and mortality as judged by the evaluation of the clinical score of the positive individuals (especially Gamma and Delta VOCs). This was observed especially during the Gamma and Delta VOC waves. In that line, during the Gamma VOC wave, nearby cities (São José do Rio Preto city, situated about ~200 km from the town of Serrana) experienced a higher mortality rate [26], especially among young unvaccinated individuals. We believe that reduced lethality in Serrana was probably due to the anticipated vaccination program, where, by this time, 80.1% of the adult population was vaccinated. The beneficial effects of the vaccination were also observed in other studies where fully vaccinated individuals were less likely to acquire symptomatic or asymptomatic infections [27]. Therefore, the COVID-19 vaccination shows beneficial effects in reducing the SARS-CoV-2-related infection rates, severe cases, and mortality. 

Despite the highly immune population in the studied location, we observed an extremely high circulation of the Omicron VOC that was responsible for the third epidemic wave (the first cases were detected in the 51st epidemic week). The dissemination of the Omicron VOC was very high, and in a period of 3–4 weeks, Omicron VOC was a fully dominant strain in Serrana town. The Omicron VOC dissemination demonstrated that the currently applied vaccination schemes could not contain this variant. The Omicron VOC has a specific mutation profile that confers immune evasion from the commonly applied SARS-CoV-2 vaccines [28,29]. Additionally, the inactivated SARS-CoV-2 vaccines, which were widely applied in Serrana town, lost their neutralizing capacity in regard to the Omicron VOC, where only 40% of the immune serum showed low-level neutralization to Omicron with a 14.7 xfold decrease compared to the wild variant [30]. The low efficiency to Omicron VOC protection has been attributed to the m-RNA vaccines, as observed in studies performed among healthcare workers with high occupational risk of SARS-CoV-2 exposure [31]. Despite that, the vaccination could not contain the dissemination of the Omicron VOC. In Serrana, we observed very low frequency of the lethal and severe cases, similar to other locations, such as the Gauteng province in South Africa [32], and southern California, USA [33]. Moreover, beyond the Omicron VOC vaccine escape, Brazil was gradually reducing restriction measures by this time, e.g., for the holidays of children and students [34]. In addition, the Christmas period related to intensive human dispersion and traveling which might all have contributed to the rapid dissemination of Omicron VOC, not only in Brazil but also worldwide. The lower risk of severe outcomes for the Omicron infection and its interclade variation is related to the global establishment of this variation as a dominant lineage, which should alert the public health authorities worldwide [32]. 

## 5. Conclusions

In conclusion, in this study we performed longitudinal monitoring of the SARS-CoV-2 variants in a model vaccination city in Southeast Brazil. Despite the epidemic waves during the study period, we observed that the vaccination was crucial in reducing morbidity, severe cases, and mortality. The vaccination was not effective in regard to the Omicron VOC introduction and rapid dissemination, which is due to the specific mutation profile of this variant and the immune evasion of the currently used vaccines. The performed genomic surveillance plays an important role in the monitoring of SARS-CoV-2 variants, as it allows for the quick detection of new variants and monitoring of their dissemination and substitution rates. In that line, we still continue the genomic monitoring of the SARS-CoV-2 variants in this region and turn our attention to the diversification of Omicron and its sublineages that can rise.

## Figures and Tables

**Figure 1 viruses-14-02148-f001:**
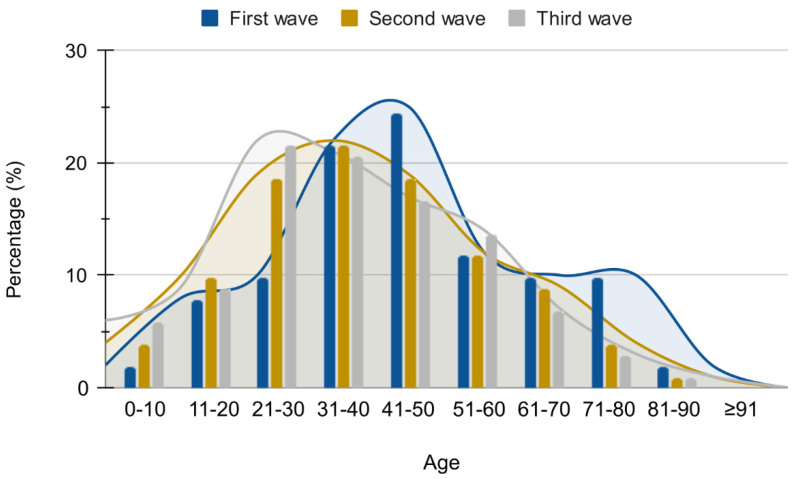
Distribution of patients according to age during the three SARS-CoV-2 epidemic waves. The X-axis contains the intervals of ages, including 0 until ≥91 years old. The axis Y represents the percentage of individuals positive for SARS-CoV-2 during each epidemic wave.

**Figure 2 viruses-14-02148-f002:**
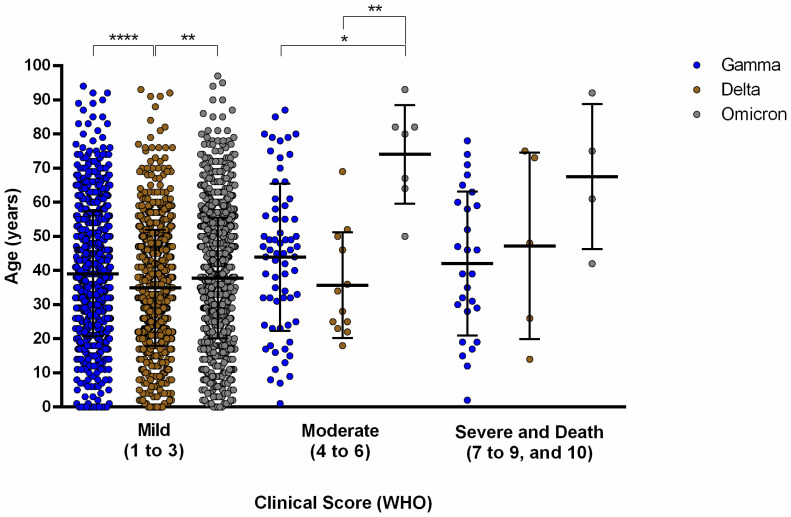
Distribution of positive COVID-19 cases caused by Gamma, Delta, and Omicron variants according to the clinical scores of the World Health Organization compared to patients’ age. Clinal scores of 1 to 3 were classified as mild, 4 to 6 as moderate, 7 to 9 were severe, and 10 for death. Data are expressed in average age (years) ± standard deviation. Statistical analysis was performed using Kruskal–Wallis with Dunn’s test for correction of multiple comparisons and α = 0.05, *p* > 0.05 (not significant), *p* < 0.05 (*), *p* < 0.01 (**), and *p* < 0.0001 (****).

**Figure 3 viruses-14-02148-f003:**
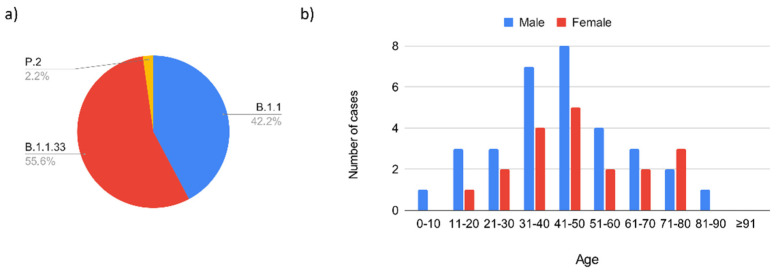
Surveillance of SARS-CoV-2 variants and clinical symptoms during the first wave. The data are distributed according to (**a**) variants of interest and concern during the first wave; (**b**) demographic characteristics of the positive patients divided by sex and age.

**Figure 4 viruses-14-02148-f004:**
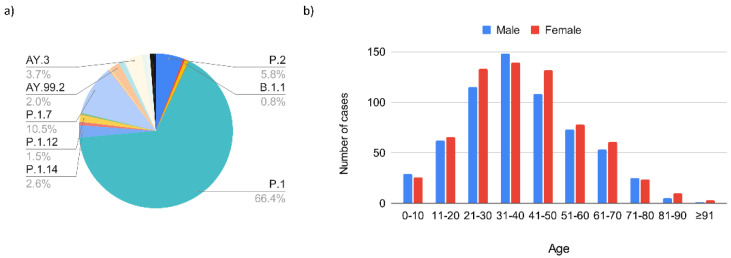
Surveillance of SARS-CoV-2 variants and clinical symptoms during the second wave. The data are distributed according to: (**a**) variants of interest and concern during the second wave; (**b**) demographic characteristics of the positive patients divided by sex and age.

**Figure 5 viruses-14-02148-f005:**
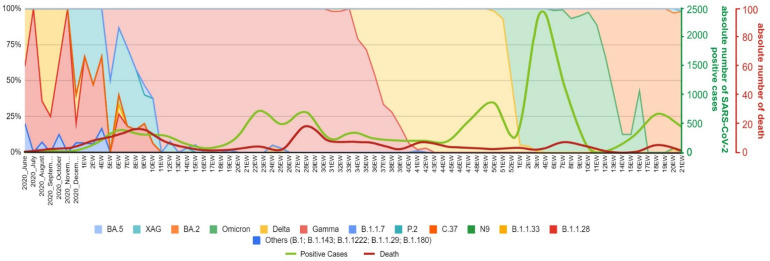
The SARS-CoV-2 epidemic in Serrana. Substitution dynamics of SARS-CoV-2 lineages by wave and epidemiological week (EW), Y-axis, frequency (%) of SARS-CoV-2 lineages by period (X-axis). Absolute number of SARS-CoV-2 positive cases ranging from 0–2500 (green line) and mortality ranging 0–20 (red line) during this period.

**Figure 6 viruses-14-02148-f006:**
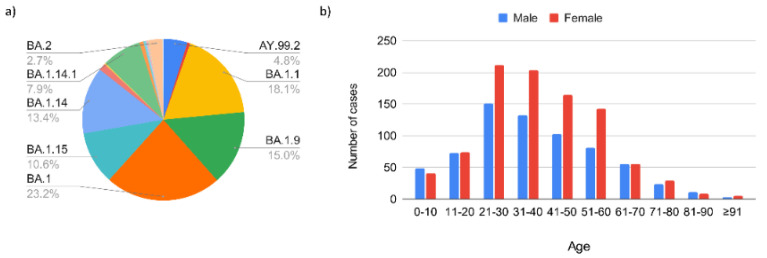
Epidemiological evaluation of SARS-CoV-2 during the Omicron epidemic wave. The data are distributed according to: (**a**) Omicron sublineages; (**b**) demographic characteristics of the positive patients divided by sex and age.

**Figure 7 viruses-14-02148-f007:**
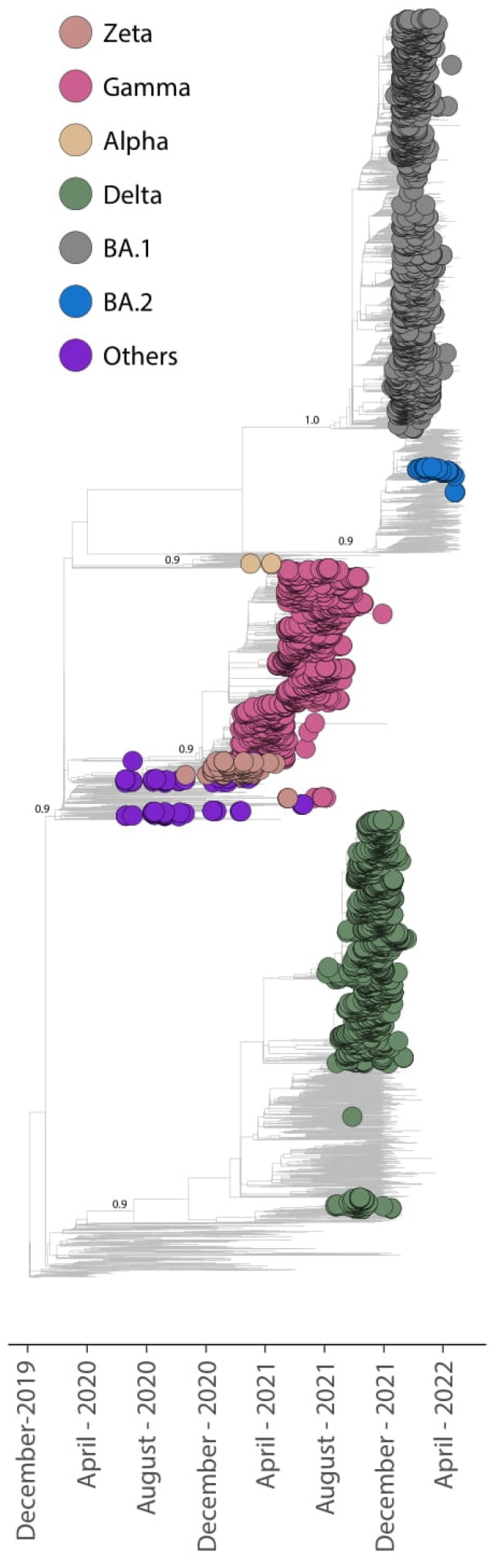
Time-resolved maximum likelihood tree containing (n = 4375) high-quality complete genome sequences from Serrana obtained in this study analyzed against a backdrop of global reference sequences. The samples obtained in this study are highlighted on the phylogenetic tree.

## Data Availability

The genomic sequences presented in this study are openly available in the GISAID public database (available online: https://www.gisaid.org/). The GISAID IDs and related metadata presented in this study are available in the Appendix A.

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
