# Peer review of "Dynamics of SARS-CoV-2 Variants of Concern in Vaccination Model City in the State of Sao Paulo, Brazil"

_viruses, 2022, doi:10.3390/v14102148_

Round 1

Reviewer 1 Report

Geneal feed back

This study investigates the circulation of SARS-CoV-2 variants  between June 2020 and March 2022, before and after the mass vaccination program against COVID-19, using a large-scale sequencing program on all real-time SARS-CoV-2 positive samples from the Brazilian city of Serrana, west of San Paolo, whose popoulation counts 46,166 inhabitants.

This study provides a description of circuating variants of concers. During the entire study period period a total 4,538 SARS-CoV-2 genomes were obtained from naso-pharyngeal swabs of COVID-19 patients and analysed. 

Whilst Delta mild cases were mostly found in the patients under 40 years,  moderate omicron cases were more likely in elderly patients.

During the first epidemic wave (February 2020 - November 2020) most COVID-19 patients were 41 - 50 old and generally presented mild symptoms (54.5%), 36.4%  moderate symptoms and in 9.1% severe disease. 

During the second epidemic wave (November 2020 - October 2021) the emergence and spread of Gamma (from February 2021) and Delta (from August 2021) variants was observed. Patients during the second epidemic wave were ounger (31 - 41 years of age) and the proportion of those affected by mild disease was much higher (89.8%).

With the third epidemic wave (December 2021 - April 2022) , which corresponds to the omicron transmission period, an exponential surge in the number of cases was observed.

Specific comments

This study drawed sensible conclusions on vaccine effectiveness for Sinovac, the main vaccine used in Brazil. However similar findings were reported also for m-RNA vaccines (mainly Comirnaty and Spikevax) used in Europe and Nord America. For instance, these 2 studies could be cited to back the concept of high vaccine effectiveness during Gamma as well as Delta transmission period: PMID: 36016081 and PMID: 35215930.

and the following study could be cited to discuss high rate of breakthrough infections in population with high vaccine coverage as health care workers during Omicron transmission time: PMID: 36016284.

Furthermore I wonder whether some molecular epidemiology analysis could be run to estimate risk factors (in addition to age and sex) associated with the circulation of different variants,  the severity of symptoms and the risk of breakthrough infection.

Author Response

Dear Reviewer 1,

In response to the referees’ comments for the manuscript ID number: 1866588 entitled “ DYNAMICS OF SARS-COV-2 VARIANTS OF CONCERN IN VACCINATION MODEL CITY IN THE STATE OF SAO PAULO, BRAZIL”, we have addressed all the specific comments by adding the required information (using track changes) to the manuscript as described above.

Sincerely, 

Simone Kashima

Reviewer 1

Comment #1: This study drawed sensible conclusions on vaccine effectiveness for Sinovac, the main vaccine used in Brazil. However similar findings were reported also for m-RNA vaccines (mainly Comirnaty and Spikevax) used in Europe and North America. For instance, these 2 studies could be cited to back the concept of high vaccine effectiveness during Gamma as well as Delta transmission period: PMID: 36016081 and PMID: 35215930.

Response #1: 

We are grateful for the valuable comment of Reviewer #1.

The major information of this manuscript is the molecular epidemiology analysis of circulating SARS-CoV-2 lineages in a model vaccination city of Brazil, where for the first time Coronavac (Sinofarm) vaccine was applied. We highlight that the study to evaluate the effectiveness of this vaccine is presented in the following article by Ferreira et al., 2022. The above suggested articles, PMID: 36016081 and PMID: 35215930 describe breakthrough SARS-CoV-2 infections in European cohorts of healthcare workers. The most important conclusions included that the complete vaccination scheme was related to less probability of acquirement of asymptomatic or symptomatic SARS-CoV-2 infection. Such a reduction of the overall morbidity due to the application of the vaccine was also observed in our study especially during the first gamma VOC wave when the lethality was much lower compared to the rest of the country. For this reason, we agree with the reviewer and added the following citation: Basso P. et al., 2022 in the “Discussion” section:

Lines 334-336, Page 10 “... . The beneficial effects of the vaccination was also observed in other studies where ful-ly-vaccinated individuals were less likely to acquire symptomatic or asymptomatic in-fection [27]...”

Comment #2: and the following study could be cited to discuss high rate of breakthrough infections in population with high vaccine coverage as health care workers during Omicron transmission time: PMID: 36016284.

Response #2: 

We are grateful to the valuable comment of the reviewer. We are aware that the high number of cases was due to the significant number of mutations present in the Omicron VOC that enabled vaccine breakthroughs and wide dissemination of this variant of concern. We decided to cite the suggested article. We performed the following modification in the discussion section: 

Lines 350-352, Page 11 “...Low efficiency to Omicron VOC protection has been attributed to the m-RNA vaccines as observed in studies performed among healthcare workers with high occupational risk of SARS-CoV-2 exposure [31]. …” 

Comment #3: Furthermore I wonder whether some molecular epidemiology analysis could be run to estimate risk factors (in addition to age and sex) associated with the circulation of different variants, the severity of symptoms and the risk of breakthrough infection.

Response #3: In this study we evaluated the SARS-CoV-2 lineage substitution in a small city that was a model example for SARS-CoV-2 vaccination in Brazil. Our objective was to analyze only the circulating SARS-CoV-2 variants among vaccinated populations rather than estimate risk factors and associations with severe disease. To contextualize better our objectives we removed the phrases of the text where we explained clinical symptology. Similar we removed the figures where this was presented. The following modifications were performed in the manuscript:

Lines 175-177, Page 5: We removed the following phrase “... The patients reported the following main symptoms: sore throat, myalgia, anosmia, dry cough, nausea and headache (Figure 3c)....”

We removed Figure 3c.

Lines 226-227, Page 6, We removed the following phrase “... The main symptoms related before and after the vaccination campaign were similar and included: coryza, muscle pain, dry cough, nasal congestion, and headache (Figure Figures 4b and 4c). …”

We also removed Figure 4c.

Reviewer 2 Report

In this study, Slavov et al. aimed to evaluate the vaccination effect on the SARS-CoV-2 dynamics and clinical outcome in the town of Serrana. They report a significant reduction of morbidity and mortality in Serrana compared to the rest of Brazil, after the end of the vaccination campaign. Authors also revealed that multiple introduction events have occurred during SARS-CoV-2 epidemic waves, and that vaccination campaign in Serrana did not prevent dissemination, neither morbidity nor mortality of Omicon variant.

The manuscript is generally well written, even if with some typos, and the authors presented a remarkable amount of clinical and genomic data. However there are some major concerns that should be addressed before manuscript publication.

In the main title and in the abstract the authors correctly underlined the importance in analyze genomic dynamic of SARS-CoV-2 during the vaccination campaign in a small community, but they did not report any data about the vaccination campaign. Authors should report in detail how the vaccination program was executed: vaccinated people were chosen randomly? Did the campaign followed an established timeline? Elder or more vulnerable people had been vaccinated with some priority? It would be interesting to better understand how vaccination campaign had influenced dissemination of SARS-CoV-2 and  mortality rate during the second wave, calculating correlation between vaccination status and VOC and/or age and/or sex. Moreover, some more data should be reported to better show which were the differences and the similarities of SARS-CoV-2 dissemination in Serrana and in the rest of Brasil.

Major concerns:

·         Figure 1: it would be useful to show the distribution of age during the three epidemic waves in the whole Brazilian country.

·         Figure 2: this is a very interesting figure that shows significative differences between epidemic waves, but I cannot find any comment of this figure in the manuscript. Please comment these data in the results and in the discussion.

·         Line 208: how vaccination campaign was organized?

·         Line 210: how did you divide the city in “groups”?

·    Line 211: please show in a graphic the timeline of: i) number of vaccinated individual, ii) rate of gamma VOC cases, iii) rate of delta VOC cases, iii) rate of severe cases. It is not clear to me to understand if could be some correlation between VOC dissemination and number of vaccinated individuals.

·         Line 250-251: are you able to guess why did you find differences in sex and age incidence, respect to the other waves?

·         Figure 7: the use of different colors to tag different VOCs is redundant as they are already recognizable because located on separated branches. Instead, please indicate major lineages with sidebars and use colors to highlight and distinguish sequences from different epidemic waves.

·         Line 268: starting from 4375 genomes, some more details on the molecular epidemiology may be addressed. For example: 1) how many alpha/gamma/delta/omicron sequences have been sequenced ? How many delta? P.2 sequences during the first and second waves were from different introductions? Delta sequences during the second and third waves were from different introductions? During October-December 2021 in Serrana you observed an increasing number of (presumably delta) cases: it is possible to guess they started from a single introduction event?

·         Line 323 and 325: how did you calculate the “dissemination rate”? Please report the related data in results.

·         Line 351: The figure S1 is not available and it is not mentioned in the text. Please add it and mention it during the text, or remove this sentence.

Minor concerns:

·         Line 112: Nextrain SARS-CoV-2 workflow comprehends many passages, please explain better how you chose and analyze 4000 genomes.

·         Lines 118 and 266: please note that you cannot realize a robust statistical support for bootstrap values of a phylogenetic tree obtained with only 1000 replicates, when you analyze thousands of sequences.  Please rephrase the sentence at line 266.

·         Line 134: Did you choose 4375 sequences over 4538 because of their high coverage length and coverage depth? If so, please rephrase the sentence and correct typos.

·         Line 239: did you mean “dislocation” (instead of “desolocation”)? Please explain better.

·         Line 242: which is the “Figure 1A” ?

·         Line 254-255: I cannot understand this sentence, please rephrase.

·         Line 264: how many reference sequences did you analyzed?  4000 as described in methods (line 118),  or 2908 ?

·         Line 319: how did you calculate the reduction in infection rate due to vaccination? Please explain better or modify the sentence.

·         Line 322: substitute the word “intensity” with “circulation”.

·         Line 346: the word “genomic” was repeated two times, please remove one of those.

Author Response

Dear Reviewer 2,

In response to the referees’ comments for the manuscript ID number: 1866588 entitled “ DYNAMICS OF SARS-COV-2 VARIANTS OF CONCERN IN VACCINATION MODEL CITY IN THE STATE OF SAO PAULO, BRAZIL”, we have addressed all the specific comments by adding the required information (using track changes) to the manuscript as described above.

Sincerely, 

Simone Kashima

Reviewer 2

Major concerns

Figure 1: it would be useful to show the distribution of age during the three epidemic waves in the whole Brazilian country.

We have demonstrated the age of the studied groups that we obtained from our proper metadata i.e. the system of the city of Serrana. As it is very time and computer consuming the estimation of the age distribution of each Brazilian state (27 states, three years of pandemic, population over 200 million) we opted to keep the same figure regarding the city of Serrana. 

Figure 2: this is a very interesting figure that shows significative differences between epidemic waves, but I cannot find any comment of this figure in the manuscript. Please comment these data in the results and in the discussion.

We are sorry that we have not mentioned the explanation of this figure in the text. We performed the following modifications:

We presented figure 2 in the following lines (Lines 145-150, Page 4): “...Among the identified VOCs and VOIs, so far, we identified 52 SARS-CoV-2 sub lineages in the studied location. The clinical outcome of COVID-19 in the tested patients was evaluated using the WHO clinical score (Figure 2). Most cases were characterized as mild (88.9, 98.1, 99.1 % to gamma, delta, and omicron, respectively) regardless of the VOC in question. Delta VOC mild cases were mostly identified in the population under 40 years of age (Figure 2). …”

Line 208: how vaccination campaign was organized?

Our objective was uniquely to examine the substitution rates of the SARS-CoV-2 variants in this city, where the Brazilian vaccination campaign. The organization of this vaccination was an objective of another study by Ferreira et al., 2022. We however performed modifications to better understand of the mass vaccination program; the paragraph was substituted by the following text: 

Lines 213-218, Page 6: “However, the second epidemic wave coincided with the mass vaccination campaign in Serrana town that started with the first dosis (on February 17, 2021) followed by the second dose 4 weeks after. From the first gamma VOC case (epiweek 5) to the complete vaccination of all individuals in the city (epiweek 15), mild cases corresponded to 71.4%, 19.1% were moderate and 9.5% were classified as severe.”

We also added the following phrase in the discussion section (line 302, page10):

“... (for more information of the campaign organization refers to reference [20])...”

Line 210: how did you divide the city in “groups”?

The division of the city in groups was performed to more easily access the population for vaccination and to organize the vaccination process. However, this was not an objective of our study, and we removed this from the text. For more details of how the vaccination campaign was performed, please refer to the following article: Ferreira, et al., 2022.

We excluded the following text from the manuscript: “... From the first gamma VOC case (epiweek 5) to the complete vaccination of all groups in the city (epiweek 15), mild cases corresponded to 71.4%, 19.1% were moderate and 9.5% were classified as severe…”

Line 211: please show in a graphic the timeline of: i) number of vaccinated individual, ii) rate of gamma VOC cases, iii) rate of delta VOC cases, iii) rate of severe cases. It is not clear to me to understand if could be some correlation between VOC dissemination and number of vaccinated individuals.

We are grateful for the valuable comment of Reviewer#2. We do not have information on the progression of the vaccination in numbers in this city as to be interposed with the SARS-CoV-2 molecular data. The start vaccination was performed in January-March, but we do not have sufficient information of what were the posteriorly applied vaccines as to comply with Delta VOC wave and the number of severe cases. For example, if Coronavac vaccine was started and the rates of gamma VOC mortality were reduced we do not have information of how progressed this vaccination process. 

Line 250-251: are you able to guess why did you find differences in sex and age incidence, respect to the other waves?

We are grateful for the valuable comment of reviewer#2.

We in fact observed an increased incidence of positive cases during the Omicron wave. This was probably related to complete reduction of the restriction measures by the flexibilization of the use of face masks, the population of children and students returned presentially to the educational institutions and this coincided with the end of the year (Nataline period) related to extensive human movement throughout Brazil. An increased prevalence among children was also observed in other studies [34]. We added the following modifications in the discussion section:

Lines 342-351, Page 11 “.... Moreover, beyond the Omicron VOC vaccine escape by this time in Brazil was observed gradual reduction of the restriction measures, holidays of the children and students [34], and the Christmas period related to intensive human dislocation and traveling that all might have contributed to the rapid dissemination of Omicron VOC, not only in Brazil but also worldwide.. …”

Figure 7: the use of different colors to tag different VOCs is redundant as they are already recognizable because located on separated branches. Instead, please indicate major lineages with sidebars and use colors to highlight and distinguish sequences from different epidemic waves. 

We are grateful for the valuable comment of reviewer#2. The colors used in the presented phylogenetic tree are highlight our samples and if they are removed it will be impossible to distinguish between the tested samples and the sequence background in this study (in grey color). Therefore, we opted not to change the phylogenetic tree. 

Line 268: starting from 4 375 genomes, some more details on the molecular epidemiology may be addressed. For example: 1) how many alpha/gamma/delta/omicron sequences have been sequenced? How many delta? P.2 sequences during the first and second waves were from different introductions? Delta sequences during the second and third waves were from different introductions? During October-December 2021 in Serrana you observed an increasing number of (presumably delta) cases: it is possible to guess they started from a single introduction event?

We inserted the absolute number of sequences that were obtained from each lineage. 

We also performed an extensive search on the branches of the phylogenetic tree and observed that the Delta VOC sequences from the period between October-December were interspersed between each other. They were not related to the formation of a monophyletic cluster which is related to multiple introductions and reintroductions of delta VOC over time in this region. We added this information in the text:

Lines 139-141, Page 4: We added the percentage of each lineage regarding the total number of genomes “... The analyzed genomes were distributed by VOC like this: 1653 delta cases (37.8%), 1053 gamma cases (24.1%), 1513 omicron cases (34.6%), 75 zeta cases (1.7%) and 81 cases of other lineages (1.9%). …”

We also added the following phrases in the section “Global SARS-CoV-2 phylogenetic analysis in the town of Serrana”

Lines 275-277 Page 8 “... For example the delta cluster was represented by strains that were mixed considering their collection date which was relevant of multiple delta introduction in this region without the presence of single introduction event. …”

Line 323 and 325: how did you calculate the “dissemination rate”? Please report the related data in results.

We ate grateful for the valuable comment of reviewer #2 We did not calculate dissemination rate and therefore we removed these words from the text. 

Line 351: The figure S1 is not available and it is not mentioned in the text. Please add it and mention it during the text, or remove this sentence.

This figure does not exist and it was inserted in the text by our error. We are sorry for this and we removed this from the text.

Minor concerns:

Line 112: Nextrain SARS-CoV-2 workflow comprehends many passages, please explain better how you chose and analyze 4000 genomes.

The phylogenetic tree is composed by 7 283 genomes: 4 375 genomes from Serrana town and 2 908 global representative SARS-CoV-2 genomes [12] that were downloaded as a representative dataset from GISAID (https://gisaid.org/). Sequence alignment was per-formed using MAFFT v7.475 [13] and manually curated to remove artifacts using Aliview [14]. Maximum Likelihood (ML) phylogenetic trees were constructed using Iqtree 2 [15] with statistical support of ultrafast bootstrap applying 1 000 replicates. The nucleotide substitution model was GTR+G4+F chosen according to the BIC (Bayesian Information Criterion) statistical model. The final formatting and visualization of the phylogenetic tree were performed using the ggtree R package [16]. This was also presented in the “Materials and Methods” section (Lines 111-119, Page 3). 

Lines 118 and 266: please note that you cannot realize a robust statistical support for bootstrap values of a phylogenetic tree obtained with only 1000 replicates, when you analyze thousands of sequences.  Please rephrase the sentence at line 266.

We are sorry for this error. We substituted well-supported by well-defined cluster (Lines 269, Page 8). 

Line 134: Did you choose 4375 sequences over 4538 because of their high coverage length and coverage depth? If so, please rephrase the sentence and correct typos.

We chose the sequences for this analysis based on both coverage length and depth. 

To also clarify this, we rewrote the sentence like this: Of them, we analyzed 4 375 (96.4%) sequences that presented high-quality parameters including a mean number of reads (986 784), mean coverage (98.5%) and mean depth, i.e 5 610 (Lines 136-138, Page 3-4). 

Line 239: did you mean “dislocation” (instead of “desolocation”)? Please explain better.

We switched to dislocation.

Line 242: which is the “Figure 1A” 

We corrected to Figure 1.

Line 254-255: I cannot understand this sentence, please rephrase.

Due to differences in the line numbers between the files we could not find this phrase. Please, if this manuscript goes to further revision, cite the phrase in the Reviewer letter. 

Line 264: how many reference sequences did you analyzed?  4000 as described in methods (line 118),  or 2908 ?

We selected a global reference dataset containing 2 908 sequences from GISAID. We corrected this information in the “Materials and Methods” section (Lines 112, Page 3). 

Line 319: how did you calculate the reduction in infection rate due to vaccination? Please explain better or modify the sentence.

We did not calculate the reduction in the infection rate due to vaccination in our manuscript. We correlated the application of the vaccine to the clinical score of the positive patients during the gamma wave and we observed a general reduction compared to the previous wave.  

Lines 315-317, Page 10: We changed the following phrase: “An important issue considering the vaccination program in Serrana was the effect of the vaccine reducing the intensity of the epidemic waves that was related to decreased morbidity and mortality .”

to: “... We believe that the application of the vaccine in Serrana town was related to reduced morbidity and mortality as judged by the evaluation of the clinical score of the positive individuals (especially gamma and delta VOCs). …” 

Line 322: substitute the word “intensity” with “circulation”.

The word “ intensity” was substituted by “circulation”.

Line 346: the word “genomic” was repeated two times, please remove one of those.

The word “genomic” was removed from the text.

References used in this study:

Ferreira NN, Garibaldi PMM, Moraes GR, Moura JC, Klein TM, Machado LE, Scofoni LFB, Haddad SK, Calado RT, Covas DT, Fonseca BAL, Palacios R, Conde MTRP, Borges MC. The impact of an enhanced health surveillance system for COVID-19 management in Serrana, Brazil. Public Health Pract (Oxf). 2022 Dec;4:100301.

Round 2

Reviewer 2 Report

In this study, Slavov et al. aims to evaluate the vaccination effect on the SARS-CoV-2 dynamics and clinical outcome in the town of Serrana. They report a significant reduction of morbidity and mortality in Serrana compared to the rest of Brazil, after the end of the vaccination campaign. Authors also revealed that multiple introduction events have occurred during SARS-CoV-2 epidemic waves, and that vaccination campaign in Serrana did not prevent dissemination, neither morbidity nor mortality of Omicon variant. In the current version the authors have largely resolved my previous concerns with the manuscript, and it is now suitable for publication without further revisions.